# Hand Gesture Signatures Acquisition and Processing by Means of a Novel Ultrasound System

**DOI:** 10.3390/bioengineering10010036

**Published:** 2022-12-28

**Authors:** Stefano Franceschini, Michele Ambrosanio, Vito Pascazio, Fabio Baselice

**Affiliations:** 1Department of Engineering, University of Napoli Parthenope, Centro Direzionale, 80143 Napoli, Italy; 2Department of Economics, Law, Cybersecurity and Sport Sciences, University of Napoli Parthenope, Via Medina 40, 80133 Napoli, Italy

**Keywords:** hand-gesture, person identification, neural networks, security, ultrasound system

## Abstract

Hand gestures represent a natural way to express concepts and emotions which are peculiar to each culture. Several studies exploit biometric traits, such as fingerprint, iris or face for subject identification purposes. Within this paper, a novel ultrasound system for person identification that exploits hand gestures is presented. The system works as a sonar, measuring the ultrasonic pressure waves scattered by the subject’s hand, and analysing its Doppler information. Further, several transformations for obtaining time/frequency representations of the acquired signal are computed and a deep learning detector is implemented. The proposed system is cheap, reliable, contactless and can be easily integrated with other personal identification approaches allowing different security levels. The performances are evaluated via experimental tests carried out on a group of 25 volunteers. Results are encouraging, showing the promising potential of the system.

## 1. Introduction

Biometric identification systems have proven their usefulness in several fields [1,2]. The exploitation of morphological or behavioral personal characteristics has proven to be effective for personal identification as an alternative to passwords or smart cards. The basic idea is that the recognition is carried out by analysing the subject’s peculiarities rather than an object in his/her possession (i.e., smart card, token generator) or something that he/she knows (i.e., passwords). In other words, the sought identification trait is related to the nature of the subject rather than to something in the subject’s possession or knowledge [3].

Independently from the specific recognizing trait, a biometric identification system must satisfy constraints related to the overall cost, robustness of the approach, availability and permanence of the adopted trait. Since biometric measurements have to be easy and remain stable over time, the specificity and the universality of these systems are crucial.

Nowadays, several biometric identification systems are adopted in real-world applications, including fingerprint, face, voice and iris recognition [3,4,5,6,7]. All these systems are very accurate, but also show some limitations, e.g., usage discomfort, the ease of feature replication and the complexity of the system. For these reasons, a new class of biometric traits has been recently considered, i.e., the subject’s behavioural peculiarities. The idea consists in recognizing subjects from the way they execute, intentionally or unintentionally, actions rather than their morphology [6]. Naturally, it is based on the assumption that each subject shows specific behaviours, which are different from other subjects and remain stable over time. This kind of behavioural biometric identification system has proved to be more robust to spoofing attacks compared to morphological ones [8].

In order to facilitate wide-spread adoption of behavioral recognition systems, the research community effort has focused on the identification of easily measured and detected behavioural biometrics. In [9], for example, the discrimination properties of the ECG signal are evaluated, while in [10] the brain’s electrical activity is considered. Even if both approaches show very good accuracy, they require an uncomfortable acquisition. In [11], a digital signature identification system is proposed. The idea consists in analyzing the way the signature is carried out by the subject, instead of its details. In [12] the identification problem is treated by analyzing the mouse gesture. Despite its simplicity and accuracy, the approach requires a contact system that might be a limitation for some applications. In [13], a system that implements a frequency-modulated continuous wave (FMCW) radar for recording gait details by means of Doppler spectrum analysis is presented. The main limitation consists of the wide area required for optimum functioning. The Authors of [14,15] performed personal identification by means of signals acquired through smartphone accelerometers and gyroscopes during a hand-gesture execution. Conversely, in [16] a solution based on capacitor-sensor acquisitions in which the users tap and swap different gestures on a smartphone is proposed. These approaches show good performance, but still require a smartphone with a properly installed application.

Within this manuscript, a novel identification system based on recognition of subject hand gestures is proposed. A prototype has been built and used for testing its performance. We choose to focus on hand gestures as they represent an easy and direct method of communication. Studies have shown that, during a conversation, about 55% of information is provided by attitude and gestures [17]. Even if two subjects execute the same gesture, each one will produce unique micro-movements. Thus, even the same gesture could appear different if properly processed. For example, in [18,19,20,21], personal identification prototypes based on in-air signatures or hand gesture recognition via 3D cameras or Kinect system have been presented. Moreover, in [22], a first approach of an identification system based on hand gestures signatures acquired via a Wi-Fi based framework has been proposed. Results are encouraging but the accuracy level is not enough to make the system reliable in high-risk application security.

In this article, information regarding small differences in hand gesture execution is acquired by an ultrasound prototype. The system transmits a monochromatic acoustic wave and receives the signal back-scattered by subject’s hand. The received signal will contain the information related to the movement of all hand elements (finger, palm, wrist) by means of its Doppler components. We chose a sonar system rather than a camera-based one in order to obtain a cheap solution with no privacy issues, as no image is acquired.

Sonar technologies have been already exploited for detection and discrimination of human movements. In particular, a wideband sonar for human activity classification has been proposed in [23]. The system, exploiting pulsed frequency modulated signals, is able to retrieve position and velocity of objects and discriminate static and dynamic contributions in order to distinguish, for example, standing or walking subjects. More detail on biometric systems based on ultrasound sensors can be found in [24].

In this paper, in order to obtain a cheap and easy solution, continuous wave signals without any frequency modulation (i.e., monochromatic waves) are considered. In this case, the acquired information only rely on the dynamic features, which depends on the hand’s movements instead of its shape and size. The idea is that every subject has a unique hand behavior, and thus a unique Doppler spectrum, which differs from other subjects and remains stable over time. Being a behavioral biometric measurement, it has the advantage that in cases where gesture is compromised, the user can easily change it, as in the case of a password. Another advantage is that, conversely from systems based on morphological biometrics, gestures can be concatenated to obtain more robust identification. The performance of the presented system has been tested on a real data set involving 25 volunteers.

The remainder of the paper is the following: in Section 2, the hand-gesture person identification prototype is presented, with details related to the hardware components and the software architecture. Section 3 contains a performance analysis of the proposed system and finally, in Section 4, conclusions are drawn.

## 2. System Description

This section provides a brief description of the basic theory which inspired the design of the system. Hardware components and software implementation details are also provided.

### 2.1. Proposed Methodology

Let us consider an ultrasound (US) sonar system, in which a monochromatic ($f=f_{0}$) continuous wave is transmitted. By considering a single point scatterer moving at velocity $v\left(t\right)$, the back-scattered received signal is a function of the radial component of the scatterer’s velocity $v_{r}\left(t\right)$, i.e., the velocity component in the direction of the system. Thus, it can be written as:(1)$$s\left(t\right)=A\left(t\right)\cos\left\{2\pi f_{0}\left[1+\frac{v_{r}\left(t\right)}{c}\right]t+\phi \left(t\right)\right\}\;,$$
with A(t) and ϕ(t) being the amplitude and phase of the received signal s(t), and *c* propagation speed of the mechanical wave.

In the following, we consider a scenario in which the subject hand backscatters the signal. In this case, the target can be modelled as a set of *N* non-stationary point targets, each with a specific radial velocity $v_{r,i}\left(t\right)$ and each producing an echo with different amplitude $A_{i}$, phase $\phi_{i}$ and Doppler frequency $f_{D,i}=\frac{v_{r,i}\left(t\right)}{c}f_{0}$, with $i=\left\{1,2,\ldots ,N\right\}$. Thus, the received signal can be written as [25]:(2)$$s_{tot}\left(t\right)=\sum_{i=1}^{N}A_{i}\left(t\right)\cos\left\{2\pi \left[f_{0}+f_{D,i}\left(t\right)\right]t+\phi_{i}\left(t\right)\right\}\;.$$

For the sake of clarity, in Figure 1 a brief sketch illustrates the considered scenario, showing signals back-scattered by different parts of the hand. The received signal is, thus, a coherent sum of the contributions due to all the scatterers impinged by the US wave, as reported in Equation (Equation 2). Therefore, the resulting spectrum is composed of several components, each with a proper amplitude, phase and frequency which are all time-dependent quantities. Considering the velocities of the hands, the Doppler bandwidth of the received signal is much smaller than the carrier frequency.

By manipulating Equation (Equation 2) and moving its formulation in the baseband case, the resulting backscattered signal can be written as:(3)$$s_{tot}^{bb}\left(t\right)=\sum_{i=1}^{N}A_{i}\left(t\right)\cos\left\{2\pi f_{D,i}\left(t\right)t+\phi_{i}\left(t\right)\right\}\;.$$

The baseband signal is non-stationary, thus, to provide a proper input to the CNN detector, a time/frequency transform of $s_{tot}^{bb}\left(t\right)$ is considered. Further details about this pre-processing step are provided in Section 2.2.

### 2.2. Hardware and Software Details

The system is composed of a continuous-wave ultrasonic apparatus and a software detector. The acquired signal is processed to extract its time/frequency characteristics and is analyzed by the detector in order to identify the subject. The system is composed of a waveform generator (mod. 33220A, Agilent Technologies), two US transducers (mod. 40LT16 and 40LR16, respectively, SensComp) and an analog-to-digital converter (mod. USB-6343, National Instruments). All the signals are processed in a MatLab environment (Figure 2). The adopted frequency is 40 kHz with a wavelength of approximately 8.5 mm in the air, which ensures a scattered signal, due to fingers motion of proper amplitude [26]. Such frequency proved to be effective to capture fingers movement in hand gesture recognition framework [27]. The received signal is sampled at 16 bits and 250 kHz.

The Doppler components of the acquired signals are extracted by means of the following transformations [28,29]:*Short-time Fourier Transform (STFT)*, a sequence of Fourier transforms of a windowed signal. The result is a two-dimensional (2D) signal representing both time and frequency information, and its absolute value is named *Spectrogram*.*Wavelet Transform (WT)*, which decomposes the signal in a linear combination of orthogonal functions (“*Wavelets*”). Conversely from STFT, Wavelet transform provides high frequency resolution at low frequencies and high time resolution at high frequencies. The matrix of Wavelet coefficients could be seen as a time/frequency representation named *Scalogram* or *Scaleogram*.*Stokewell Transform (ST)*, also known as *“S-Transform”*, which is an extension of the Wavelet transform using a specific Wavelet whose size is inversely proportional to the frequency of the signal.*Hilbert–Huang Transform (HHT)*, which decomposes the signal into oscillatory waves named *“Intrinsic Mode Functions*” (IMF), which are characterized by a time-varying amplitude and frequency. Subsequently, the instantaneous frequencies of each IMF are computed, obtaining the so-called *Hilbert spectrum*.*Constant Q Transform (CQT)*, which follows similar procedures with respect to the STFT with the main difference of non-uniform frequency resolution of time-frequency representation.

Figure 3 shows the first IMF of the received signal in the case of two different subjects performing the same gesture, i.e., hand closing. It is possible to note that, although the general trend of the signal is quite similar, there are visible differences due to some peculiarity in the gesture execution. This is what we are interested in for the identification purpose. The processing chain of the proposed system was developed in order to detect such differences in order to identify a particular user even if his/her gesture is copied by an hypothetical impostor. Further, a deep learning approach based on convolutional neural network (CNN) was adopted for the detection. This kind of architecture is widely used in classification, segmentation and regression problems, proving to be very flexible [30,31,32]. The employed CNN topology consists of three 2D 5×5 convolutional layers with 32 filters and a rectified linear unit (ReLU) after each layer. Mini-batch size was equal to 64 and Adam algorithm, with an initial learning rate of $10^{-4}$, has been exploited as optimizer. In order to reach the convergence, the network has been trained for 100 epochs. The cross-entropy was adopted as cost function in the training process. In detail, in the case of a training set composed of *N* entries, each one consisting of the acquired data *X* and the related label (*g* and $\bar{g}$ in case of *genuine* and *impostor* subjects, respectively), the cross-entropy function L(·) can be written as [33]:(4)$$\begin{array}{cc} L(X,\Phi )= & -\frac{1}{2N}\sum_{n=1}^{N}\{p_{ref}\left(g|X_{n}\right)ln\left[p_{net}\left(g|(X_{n},\Phi )\right)\right]+\\ \; & +p_{ref}\left(\bar{g}|X_{n}\right)ln\left[p_{net}\left(\bar{g}|\left(X_{n},\Phi \right)\right)\right]\} \end{array}$$
where Φ are the network-tuning parameters (weights and biases), $p_{ref}$ represents the true labels (genuine or impostor) of the samples, while $p_{net}$ represents a detector score related to the assignment of the considered gesture sample to genuine or impostor class.

Let us assume that the task consists in the recognition of a person in a set of possible users. After the training process, the CNN provides a membership score (*P*), i.e., a value between 0 and 1, which can be seen as the (estimated) probability that the genuine subject has been acquired: values close to 1 indicate that, for the CNN, the gesture was most probably executed by the considered user; conversely, if the membership score is close to 0, the gesture was very likely done by one of the other users. The detection is performed by comparing the score to a previously set threshold Th. By choosing a different threshold value it is possible to select different working points and characterize them by measuring the related performance. The overview of the whole processing chain is reported in Figure 4.

## 3. Prototype Performance Evaluation

In order to test the performance of the proposed system, different one-vs-all identification tests were considered. Such tests simulate a possible scenario in which an authorized subject has to be authenticated and several impostors try to cheat the system. For this purpose, a data set has been acquired by asking 25 subjects to repeat different kinds of gestures several times.

### 3.1. Acquisition Protocol

The volunteers have been asked to repeat the same three gestures placing the hand in front of the prototype, approximately 30 cm distant. The considered three gestures are the hand *closing and opening* (Gesture 1), the *shift from left to right* (Gesture 2) and the *key-tapping* (Gesture 3). Pictures of the three considered gestures are shown in Figure 5.

The acquisition campaign was divided into two different phases. In the first one, all the subjects repeated each of the three considered gestures 40 times. These repetitions were used to train the CNNs. In the second acquisition phase, which was used for testing the system performance, each user repeated the same three gestures 50 times. This phase took place with variable delays (from 30 min to one week) from the first one. The overall dataset is composed of 2250 samples per each gesture, of which 1000 are used for training the detector and the remaining 1250 are used to test the prototype and produce the results of Figure 6, Figure 7, Figure 8 and Figure 9 and Table 1 and Table 2.

In this framework, an augmentation technique has been adopted in order to increase the systems performance. In particular, every gesture sample used for the detector training has been temporally shifted in order to obtain five different versions. In this way, the CNN detector has been trained to recognize the Doppler gesture signatures independently from their position in the observation window.

It is worth noting that we assumed all the users, and thus also the impostors, knew the gestures used for the identification perfectly. This is a strong assumption that made us consider the worst case scenario.

### 3.2. Results

In order to evaluate the discrimination performance, an identification test was performed considering as genuine a single user per time. Assuming that the subject to be identified is labeled as *genuine*, while the others are named *impostors*, for each working point (i.e., Th value) the *True Positive Rate* (TPR), defined as the ratio between the correct *genuine* assignment against all the *genuine* testing samples and the *False Positive Rate* (FPR), that is the ratio between the incorrect *genuine* assignment against all the *impostor* gesture samples, are computed. It is also worth recalling their complementary quantities, the *False Negative Rate* (FNR), defined as the ratio between the incorrect *impostor* assignment against all the *genuine* sample and the *True Negative Rate* (TNR), that is, the ratio between the correct *impostor* assignment against all the *impostor* samples.

Based on TPR and FPR values, it is possible to compute the Receiver Operative Characteristic (ROC) curve [3] for each subject and identify the optimal pre-processing algorithm and gesture type. From each ROC curve two synthetic metrics were extracted; the Area Under ROC (AUR), representing the overall performance of the receiver, the closer this value is to 1 the better the receiver has to be considered; the other metric is the Equal Error Rate (EER) defined as the threshold which gives an FPR equal to FNR. In this case, values closer to 0 represent better performance.

An important preliminary step is to identify the most promising kind of pre-processing for the considered application. The AUR and EER values reported in Figure 6 and Figure 7 show that the system trained with the Hilbert–Huang Transform of the received signals has the worst performance while the STFT the best one, both in terms of median value and 75th–25th percentiles differences. For this reason, in the following analyses only the STFT case is reported. The second analysis is aimed at identifying which is the best gesture candidate for the considered test case. For this purpose, the average ROC curves for each of the tree gestures are shown in Figure 8, while the TPR values for different FPR, AUR and EER values are summarized in Table 1. From these results it is evident that the choice of the gesture has a significant impact on the prototype performance; in particular, gestures 1 (closing and opening) and 2 (shift from left to right) help the system in the enhancement of differences between the subjects. This is probably due to their Doppler spectra richness, produced by the faster and wider movements they have with respect to gesture 3 (key tapping).

Furthermore, a third analysis was performed on the evaluation of the time reliability, i.e., the performance stability over time, which is an important factor for an identification system. This problem generally affects systems based on behavioural biometrics. If the observed trait remains the same and, consequently, the performance of the system remains stable, the need to periodically reacquire the specific biometric attribute is limited. A volunteer was asked to repeat the second acquisition phase twice, 10 minutes and 4 days later than the first phase, respectively. These two sets, referred to as *Short-Time Delay* and *Long-Time Delay*, produced the ROC curves reported in Figure 9 and the values reported in Table 2. A slight worsening of the performance happens, which suggests that gesture re-acquisition and CNN re-training after a certain amount of time is recommended for this system.

It is worth noting that state of the art solutions, such as fingerprint, iris, face and voice recognition systems can achieve better performance compared to the proposed system (in the actual configuration). However, the strength of the proposed approach is the possibility of being used in a multi-biometric configuration (as suggested in [34]), in order to increase the robustness of the overall system or in stand-alone configuration, where the scenario is critical from environmental point of view. In more detail, ultrasound systems are robust against light changes and acoustic noise conversely to iris and facial technologies or voice systems, and have no privacy issues.

### 3.3. Comparison with the State of the Art

As seen in Section 3.2, the proposed US hand gesture identification system is able to reach, in average, an EER equal to 13% when gesture 1 is considered. In this subsection, a comparison between the identification performance of the proposed system and other state-of-the-art solutions is provided.

In detail, in [12] a person identification system based on mouse gesture is considered. The system achieves an EER of 5.11% for a test involving 39 subjects (29 genuine users and 10 impostors). This EER value was obtained for a combination of four different gestures (single-gesture results were worse).

Moreover, [19] proposes a system based on an in-air signature verification system. The system is mainly composed of a depth camera for the acquisition step and a deep learning detector. In this work, 5.5% of EER was achieved for a test involving 40 subjects in a one-versus-five authentication scenario.

Finally, an identification system based on hand gestures acquired by a 3D camera is proposed in [20]. A forgery testing involving six subjects showed that the system EER was approximately 26% in a one-versus-five identification scenario.

### 3.4. Computational Burden

As a final analysis, the computational burden of the proposed approach is measured. In Table 3 a comparison of processing times for the different considered time-frequency representations of the prototype is reported. In particular, the *Training Time* is the time needed for the training of the CNN, while the *Testing Time* is the time needed to make the prediction after the CNN has been trained.

It is worth noting that the training time depends on the number of epochs needed by the CNN for the convergence. For this operation, several minutes (between 6 and 70) are required; thus, it has to be done offline. Conversely, the testing time is essentially that required for obtaining the considered time/frequency representation plus the time for its multiplication with the CNN weights. Generally, it is in the order of milliseconds, making the system capable for real time or quasi-real time applications.

From Table 3 it is possible to underline that, further obtaining the best performance, the low training/testing time of the STFT approach suggests its use in time-critical applications.

All the evaluations were performed in a MatLab environment on a Linux 64 bit workstation with an AMD Ryzen 3990X processor and an NVIDIA Quadro RTX 6000 graphics card.

## 4. Conclusions

In this work, a novel system for person identification purposes which exploits hand-gesture information acquired by means of ultrasound waves is presented. By exploiting the Doppler signature of scattered pressure waves, the prototype is able to extract a representative pattern for each user and to use it for identification. The Doppler information is extracted via appropriate time/frequency transformations, while the detection is performed through convolutional neural networks. A prototype was built for testing the performance of the system on real data. Due to the ultrasound technology and the low computational complexity, the prototype is compact, light and cheap. The experimental results, in one-vs-all person identification scenarios, showed that the proposed solution seems very promising due to good identification performance and low computational time. Its compactness and robustness to environmental noise allow the adoption of the proposed system in different scenarios, both in a multi-biometric system (combined with other systems such as fingerprint, voice, facial recognition, etc.) as well as in stand-alone configurations. However, similar to other behavioral biometric systems, problems related to time stability and background clutter may occur. Therefore, future development will be focused on improving system stability and robustness. 

## Figures and Tables

**Figure 1 bioengineering-10-00036-f001:**
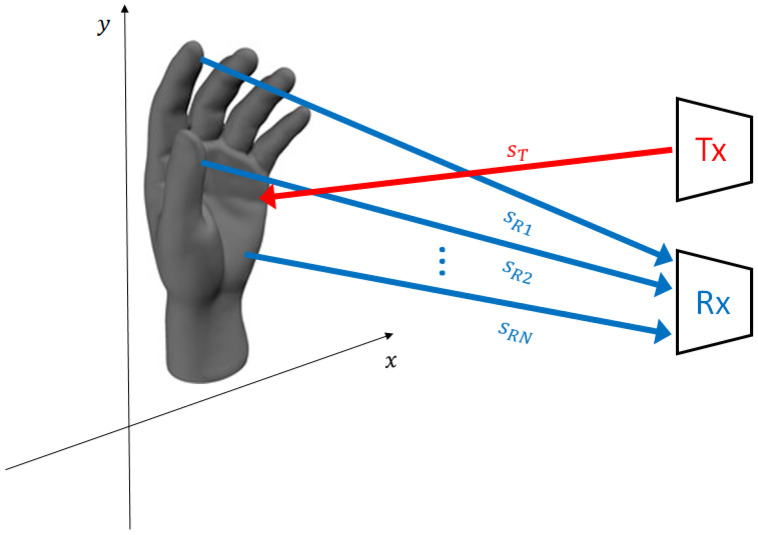
The signal back-scattered by the hand is an ensemble of multiple waves reflected by the different parts of the hand. Each one is characterized by its own amplitude and phase and, in case of a moving hand, Doppler frequency.

**Figure 2 bioengineering-10-00036-f002:**
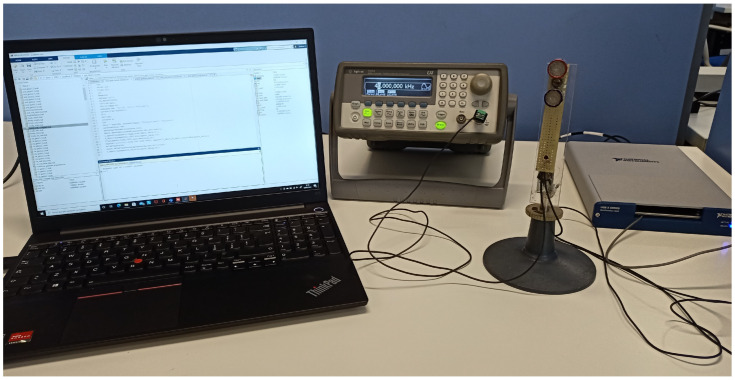
Picture of the proposed ultrasound-based person identification system. From left to right: a standard laptop used for signal processing, the waveform generator, the ultrasound transducers and the analog-to-digital converter.

**Figure 3 bioengineering-10-00036-f003:**
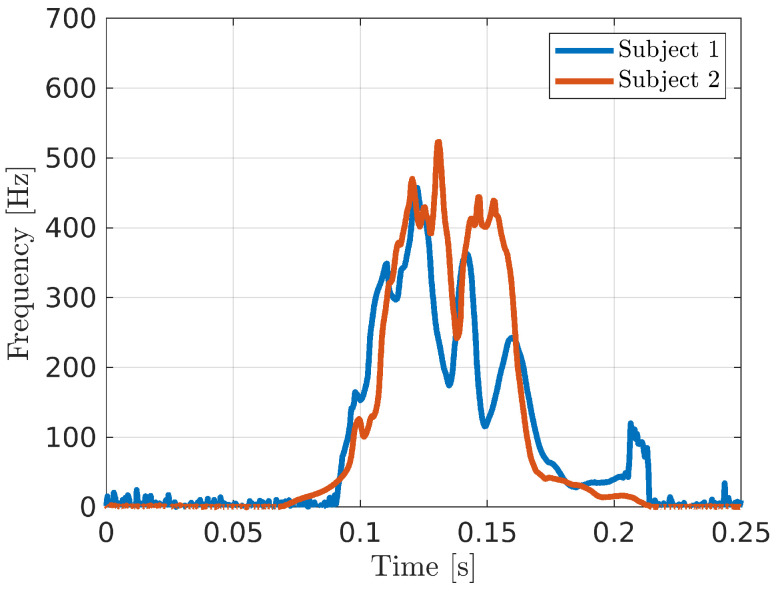
Instantaneous frequency of the first IMF of the received signal in case of two subjects executing the same gesture.

**Figure 4 bioengineering-10-00036-f004:**
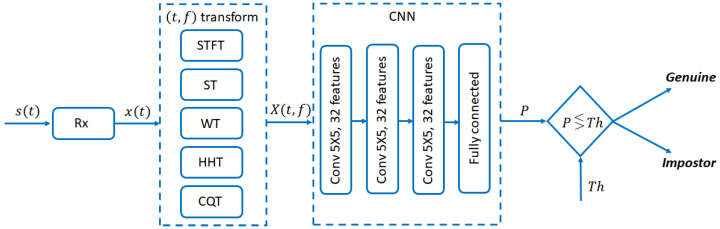
Processing chain of the proposed system. The acquired signal s(t) is demodulated and filtered by the receiver Rx, obtaining the base-band signal x(t). X(t,f) is subsequently computed by means of a time/frequency transformation and provided as input to the CNN. Its output, i.e., the membership score *P*, is compared to the threshold Th in order to obtain the genuine/impostor detection.

**Figure 5 bioengineering-10-00036-f005:**
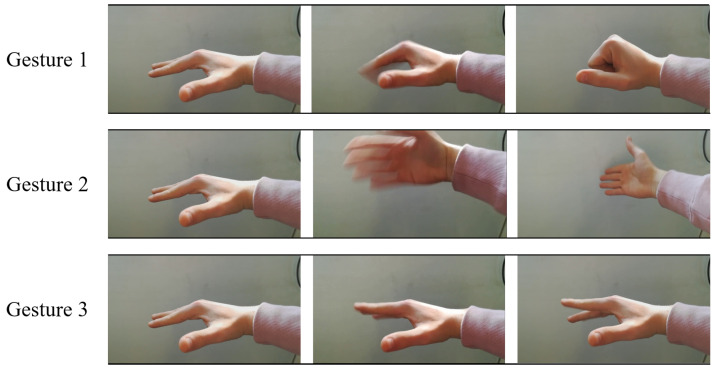
Picture of the considered gestures: *hand closing and opening* (Gesture 1), *hand shift from left to right* (Gesture 2) and *key-tapping* (Gesture 3).

**Figure 6 bioengineering-10-00036-f006:**
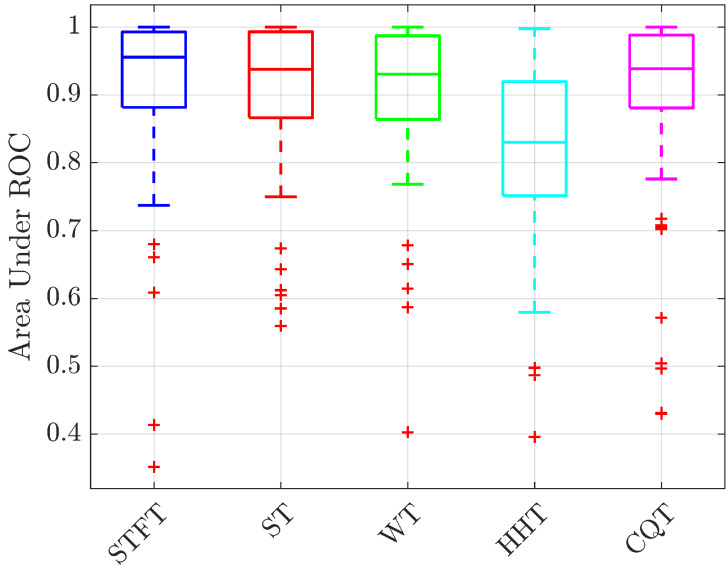
Boxplot of the Area Under ROC (AUR) for the considered pre-processing strategies. Each boxplot includes mean results for 3 gestures and 25 subjects.

**Figure 7 bioengineering-10-00036-f007:**
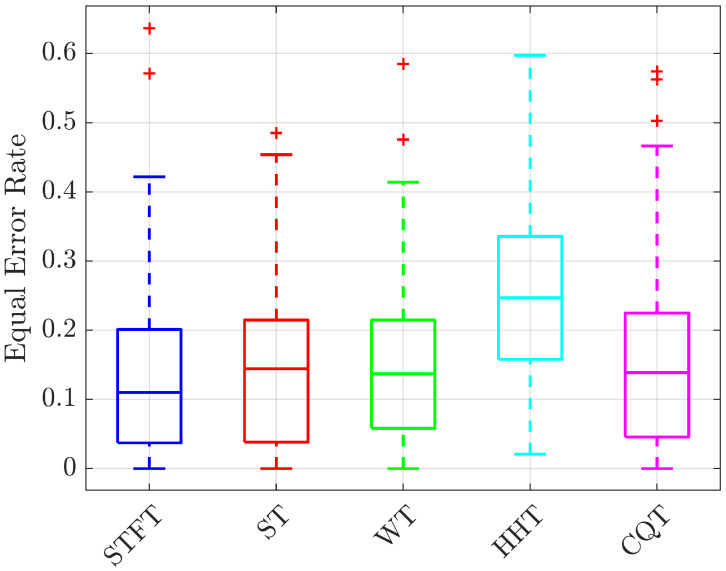
Boxplot of the Equal Error Rate (EER) for the considered pre-processing strategies. Each boxplot includes mean results for 3 gestures and 25 subjects.

**Figure 8 bioengineering-10-00036-f008:**
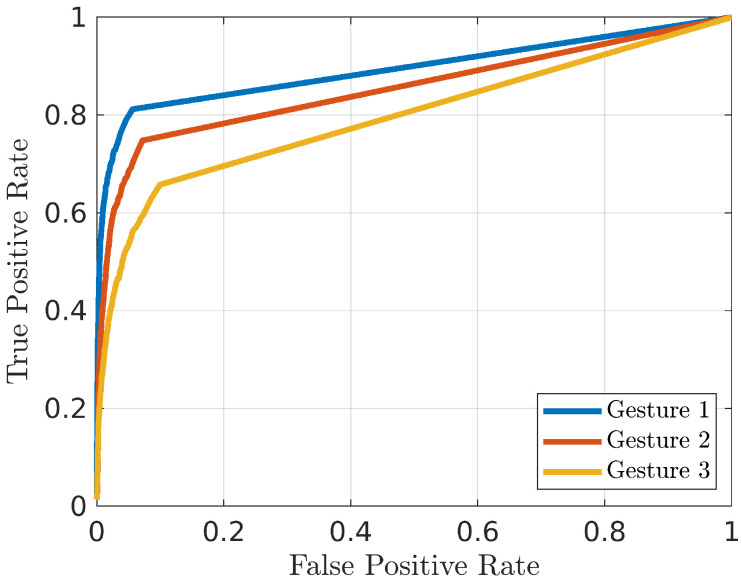
ROC of the three considered gestures. Each curve is averaged on the 25 subjects.

**Figure 9 bioengineering-10-00036-f009:**
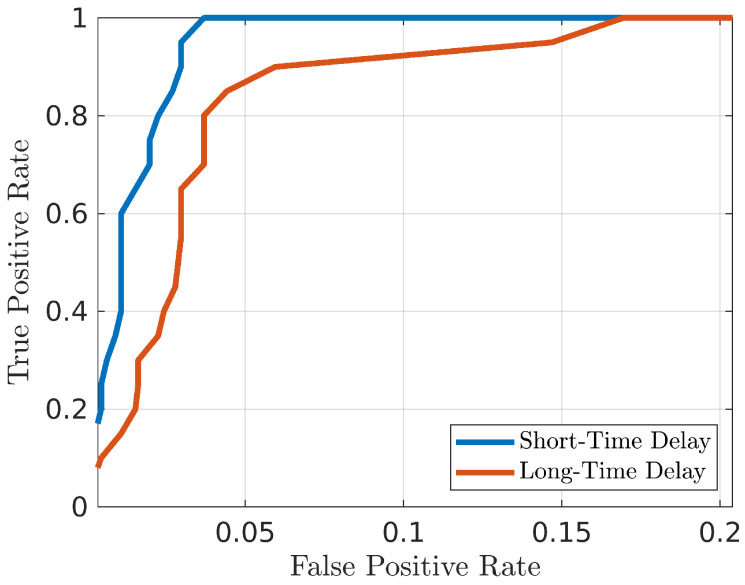
ROC curves in case of 30 min delay (Short-Time Delay) and 4-day delays (Long-Time Delay) between training and testing acquisitions.

**Table 1 bioengineering-10-00036-t001:** Performance comparison (TPR, AUR and EER) for the three considered gestures.

	TPR for FPR = 0.005	TPR for FPR = 0.01	TPR for FPR = 0.05	TPR for FPR = 0.1	AUR	EER
**Gesture 1**	0.51	0.61	0.8	0.85	0.92	0.13
**Gesture 2**	0.32	0.41	0.68	0.78	0.89	0.15
**Gesture 3**	0.21	0.28	0.54	0.66	0.85	0.20

**Table 2 bioengineering-10-00036-t002:** Performance comparison for testing gestures executed with different delays from the training gestures.

	TPR for FPR = 0.005	TPR for FPR = 0.01	TPR for FPR = 0.05	TPR for FPR = 0.1	AUR	EER
**Short-Time Delay**	0.29	0.4	1	1	0.99	0.04
**Long-Time Delay**	0.12	0.15	0.88	0.93	0.97	0.08

**Table 3 bioengineering-10-00036-t003:** Computational burden of the proposed approach.

	Training Time	Testing Time
**Short-Time Fourier Transform**	6 min	3 ms
**Stokewell Transform**	35 min	27 ms
**Wavelet Transform**	30 min	25 ms
**Hilbert-Huang Transform**	70 min	13 ms
**Constant Q Transform**	8 min	8 ms

## Data Availability

All the data are available prior request to the authors.

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
