# Peer review of "Hand Gesture Signatures Acquisition and Processing by Means of a Novel Ultrasound System"

_bioengineering, 2022, doi:10.3390/bioengineering10010036_

Round 1

Reviewer 1 Report

1. In ROC curve figure and explanation were the optimal threshold calculated? or was the threshold 0.5

2. 2250 samples per gestures were used where 1250 were used for test set and 1000 for training. Training samples are to small augmentation methods maybe used to get better performance.   

Reviewer 2 Report

Authors present preliminary research on hand gesture recognition as biometric identification using an ultrasound system plus related processing method, where no image is acquired thus having no privacy issues, which is kind of interesting.

Paper is not immune from some flaws, authors are required to make clarification or improvements:

1/Fig.5 shows 3 hand gestures used in this research, however, for readers it is not clear where the ultrasonic apparatus was placed, the ultrasonic device stands in front of the hand or lays beneath the hand ?

2/ it is not clear what Gesture 1-3 refer in Table 1, clarification is needed in body text or in table.

3/what’s the limitation of the proposed system, authors are required to explicitly make a discussion in “Conlusion”.

Reviewer 3 Report

The authors proposed a method for hand gesture signature recognition using ultra sound system. There are several points that the authors should address: 

·       -The authors stated that  “The adopted frequency is 40 kHz with a wavelength of approximately 8.5 mm in the air, which proved to be effective to capture fingers movement

   there is no justification why this frequency is effective?

·       Is the database publicly available? If not, authors are strongly encouraged to include database details in order to make the work replicable

·       Comparison in section 3.3. Comparison with the state of the art is far form the proposed method. The included work [12, 19, 20] are considering different technology such as 3D camera or mouse gesture and hence comparison is not fair. More relevant work shoud be included 

·       A detailed comparison table should be included

·       Where are the network hyperparameters and the details of CNN?

Round 2

Reviewer 3 Report

The authors have addressed my previous concerns. No further comments